# Reflective Practice: Co-Creating Reflective Activities for Pharmacy Students

**DOI:** 10.3390/pharmacy10010028

**Published:** 2022-02-10

**Authors:** Kalbie Hokanson, Rene R. Breault, Cherie Lucas, Theresa L. Charrois, Theresa J. Schindel

**Affiliations:** 1Faculty of Pharmacy and Pharmaceutical Sciences, College of Health Sciences, University of Alberta, Edmonton, AB T6G 2H7, Canada; kalbie@ualberta.ca (K.H.); rbreault@ualberta.ca (R.R.B.); teri.charrois@ualberta.ca (T.L.C.); 2Graduate School of Health, University of Technology Sydney, Sydney, NSW 2007, Australia; cherie.lucas@uts.edu.au

**Keywords:** reflective practice, reflection, reflective writing, pharmacy education, co-creation, assessment

## Abstract

Reflective practice is important in pharmacy education to support skill development for decision-making, critical thinking, problem-solving, and in continuing professional development and beyond. Despite the importance of reflective practice in higher education and professional practice, reflection assignments are not universally embraced by students. This project was initiated due to recent interest in the co-creation of pharmacy curriculum by students and faculty. The purpose of this project was to develop reflection assignments for pharmacy students. The principles of the analysis, design, development, implementation, evaluation (ADDIE) instructional design framework guided the development of reflection assignment templates for three focus areas: personal development, professional development, and professional identity formation. Templates included background and definitions for these specific focus areas as well as objectives, instructions, guiding questions, assessment methods, and submission requirements. A previously tested assessment rubric was adopted for reflection assignments. Development involved target audience and expert reviews and a trial implementation was held in a year 3 patient care skills course. The co-creation process enriched the experiences of students and faculty involved in it. Future co-creation projects including groups of students, formal evaluation of outcomes, and impact on the program will further support integration of reflective practice in the pharmacy curriculum.

## 1. Introduction

Reflective practice is an integral part of the pharmacy curriculum. It supports development of skills in decision-making, critical thinking, problem-solving, and is important for continuing professional development and ongoing professional practice [1]. Reflection is defined differently depending on the philosophical tradition or conceptual framework in which this multi-faceted construct is being discussed [2,3]. Scholars have framed reflection as a mindful activity providing insights about thinking [4], a process in which experience may be explored [5] or perspectives changed [6]. According to Dewey, reflection is concerned with how thinking occurs as an active and iterative process [4]. Kolb highlights reflection as integral to learning from experience [7]. Boud et al. introduce the role of emotion in the process of reflection, which involves exploring personal experience with the purpose of new understanding and insights [5]. Mezirow advocates for critical reflection with the aim of transforming personal perspectives and effecting social change [6]. Brookfield also takes a critical approach, emphasizing conscious and unconscious aspects of the reflective process through exploring assumptions through various lenses (theory, peer, students, self) [8]. Schon introduces the concept of the reflective practitioner, bringing together theory and everyday practice [9]. Often initiated by a puzzling or surprising event, reflection on current situations (reflection-in-action) and previous experiences (reflection-on-action) brings new perspectives about a situation that may stimulate a new approach in the future [9]. Both personal knowledge and perspectives of others may be involved. Drawing on prominent writings about reflection, Nguyen et al. develop a unified definition as “the process of engaging the self in attentive, critical, exploratory and iterative interactions with one’s thoughts and actions, and their underlying conceptual frame, with a view to changing them and with a view on the change itself” [2] (p. 1182).

Reflection has long been incorporated into educational activities and assessments in the context of higher education; it can be a valuable part of a student-centred approach and a means to reinforce the importance of experience as a basis for developing knowledge and skills [10,11]. Reflection on experience facilitates construction of new knowledge and understanding, thereby supporting development for future professional roles [9,12]. In educational settings for the health professions, reflection has been shown to support development of empathy [13], confidence [14], problem-solving skills and skills for coping with uncertainty [11], and relationships with instructors [15]. In pharmacy programs specifically, reflective activities have supported development of a range of skills and abilities including critical appraisal [16], reflective thinking [1], clinical decision-making [17], connecting classroom and experiential learning [18], identifying own learning needs and self-directed learning [16,19] and maximizing learning from clinical placement experiences [20].

Various educational strategies support the development of reflective practice skills; written, verbal, and artistic approaches have been taken [15,21,22]. Reflective writing may take several forms such as essays, blogs, journals, diaries, professional development plans, and learning portfolios. Examples of verbal approaches include storytelling, podcasts, and peer discussions. Poetry, painting, drawing, sculpture, and photography have also been used. The educational strategy used depends on the intended outcome and the needs of the student [22]. Some students in healthcare and pharmacy programs perceive value in reflective activities [1,16,18], while others perceive them as challenging [1,23] or burdensome [14], questioning the value of completing as well as objectivity and transparency of grading reflections [13,24,25]. Despite the importance of and emphasis on reflective practice in higher education, students may not understand the purpose of reflection assignments nor their relevance to learning and future practice [12,16]. Research has shown that acceptance by students and their ability to engage in and benefit from reflective activities improve when educators provide structured learning experiences with information regarding expectations, clear assessment criteria, adequate guidance, and helpful feedback [10,13,16,20,22,23]. Incorporating students’ perspectives on reflection and involving them in the development of educational activities may also influence how they engage with reflective practice [20,25,26]. Considering the student perspective is essential to building reflective activities that are not seen as ineffectual by students [12,25,26]. Given the importance of reflective practice, exploring educational strategies to address these challenges with student engagement warrant exploration.

There has been growing interest in students and faculty working together to create curricula in recent years [27]. In this project, working with a third-year pharmacy student, we explored and developed reflective practice activities for pharmacy education at an entry-to-practice level. This article aims to describe our co-creation process for the development of reflection assignments for pharmacy students using the ADDIE instructional design framework.

## 2. Materials and Methods

### 2.1. Educational Setting

The Faculty of Pharmacy and Pharmaceutical Sciences, College of Health Sciences at the University of Alberta offers an entry-to-practice Doctor of Pharmacy (PharmD) program [28]. PharmD students complete a minimum of two years of pre-requisite courses prior to starting the four-year program [29]. Each year has three terms: fall, winter, and spring/summer. The first three years of the program include foundational courses in five curricular streams taught in the fall and winter terms: Pharmaceutical Sciences, Behavioural, Administrative, Social, and Evidence-based (BASE) pharmacy, Pharmacotherapy, Patient Care Skills, and Interprofessional Education. The sixth stream, the Experiential Education stream, is delivered in the spring/summer terms, beginning with years 1 and 2 (8 weeks). In-class instruction occurs in knowledge-based classes, such as pharmacology and therapeutics. There are also personal development courses incorporating ethics, communication, and self-reflection. Year 4 of the program involves Experiential Education placements (32 weeks) in a variety of practice settings (e.g., community pharmacies, hospitals, ambulatory care, pharmacy organizations) and integrated seminars where students convene in groups to discuss and explore their experiences in pharmacy practice. Students must also complete one elective course, choosing from available courses in various curricular streams, including a Research and Directed Studies course option, before or during year 3 of the program. Approximately 130 students are admitted each year.

Educational outcomes for the PharmD program include reflective practice and meta-cognitive skills, that is, the ability to reflect on knowledge, skills, and beliefs that could influence self-development and professional performance [30]. Reflection on teaching approaches (see Schindel et al. [31]) and reflection on learning (see Nagy et al. [32]) are both important. Students are introduced to the theoretical foundations of reflection in year 1 of the program and are required to complete a variety of reflective activities, such as written reflections and group discussions, in conjunction with in-class courses, practice skills laboratories, and experiential education placements. While reflective activities are included in various aspects of the programs and within curricular streams, these activities are not mapped; thus, their implementation across curricular streams or the entire four years of the program may be inconsistent and sporadic.

### 2.2. Conceptual Framework

The development, implementation, and evaluation of reflective activities for pharmacy students in this study was based on the concept of co-creation [27]. Co-creation encompasses active learning, student engagement, shared decision-making, and negotiation [33]. According to Bovill et al., co-creation occurs “when staff and students work collaboratively with one another to create components of curricula and/or pedagogical approaches” [27] (p. 196). Throughout this process, students’ perspectives and contributions are recognized and valued, and the traditional hierarchy between teacher and students is diminished [33]. In partnership with faculty, students may adopt different roles in the co-creation process such as consultant, co-researcher, co-designer, or student representative [27]. Benefits of co-creation may include a socially relevant curriculum [34], an improved learning environment [35], deeper engagement in learning and teaching activities [27], enriched meta-cognitive understanding of learning and teaching processes [35], enhanced understanding of students themselves [36], development of attributes that maximize employability post-graduation [27], and transformation through disrupting previous ways of learning or teaching, reflecting on experiences, acting in new ways, and integrating new perspectives [33]. Challenges for the learner relate to unfamiliarity with the process, role, and responsibility involved in co-creation; a power imbalance may also develop due to the necessity for assessment in university programs and lack of student involvement with implementation [35]. For teachers, challenges may arise due to changes in the control of the learning environment, perceived threats associated with change, and time required for the co-creation process [35,36].

### 2.3. Instructional Design

Principles of the analysis, design, development, implementation, evaluation (ADDIE) instructional design framework guided the development of reflection assignment templates for this co-creation project [37,38]. The first author (KH), when entering year 3 of the PharmD program, engaged with the analysis, design, and development phases of the project as a co-designer [27]. This co-creating role for one student (KH) was integrated in conjunction with a directed studies elective course completed April to June 2020. The reflective practice topic was chosen by the student from a list of possible directed study projects. It was supervised by one faculty member (TJS) and supported by the other authors (RRB, CL, TLC). The implementation and evaluation phases were completed after the conclusion of the directed studies course (RRB). The University of Alberta Research Ethics Board 2 approved the evaluation (Pro001020270).

## 3. ADDIE Framework

### 3.1. Analysis

The main consideration in the ADDIE analysis phase was to identify the needs of the target audience. A multi-faceted approach to analysis was undertaken. It began with the student (KH) becoming familiar with reflective practice and educational outcomes for the program [30], reviewing selected literature on reflective practice and related instructional approaches in pharmacy and health sciences programs [13,16,19,20,25,39,40,41,42], reviewing reflective practice assignments in PharmD course syllabi, discussing with the supervisor (TJS), and ruminating on personal experiences with reflective practice. As part of this phase, the first author (KH) documented an initial reflection:
*My interest in this project stems from my own experiences with reflection. My mother is an advocate of reflection and was a role model for me, so I had some experience with reflecting on experiences. One experience stands out for me related to an OSCE-style exam in the first year of the program. I recall experiencing intense nervousness during the exam. When I left the exam, I was unhappy with one of my answers. Usually when this happens, I am satisfied to review what I recall about questions and answers and continue on. However, in this situation, my discomfort troubled me for days following the exam. I asked myself questions such as: “Will I ever be able to learn all I need to know?” or “Is this experience going to drive my self-confidence as a student?” After discussions with my mother and time to process the event, I came to the realization that making a mistake in a safe environment is okay as a learner and was able to move beyond the experience. This outcome reinforced the value of reflection to me. As a result, my dedication to reflective practice for personal and professional growth has remained strong. My positive regard for reflective practice is not status quo among all of my classmates. I understand that a large portion of classmates dislike the process of reflection and the associated assignments. Fellow students comment that reflections are a “waste of time”, “not valuable”, “a time-filler” or “taught wrong,” which were similar sentiments to what I was reading in the literature. I also learned that these perspectives were shared by students in other programs. This project provides an opportunity for me to understand more about students’ perspectives, advocate for students, and contribute to curriculum development with my professors to support reflection and reflective practice for pharmacy students.*

Based on the ADDIE analysis phase, the student and supervisor identified strategies needed to support students’ development of reflective practice skills: reflection assignments must have a focus, students must have options for formats other than formal written reflections, students must have opportunities for sharing and engaging with peers, assignment expectations must be clear, and formative assessment and feedback must be provided.

### 3.2. Design

In the next phase, core elements of reflective assignments for year 1 of the framework were designed. The information obtained during the analysis phase was considered during the design phase. The activities consisted of creating objectives, specifying learning activities to meet the objectives, and determining how the objectives would be assessed. The first step was to develop overall objectives to address student needs and educational outcomes of the program. It was determined that pharmacy students would be able to:Define reflective practice;Describe the purpose of reflection;Engage in different types of reflective activities;Explore reflective activities individually and with peers;Respond to formative and summative assessments;Identify changes for future learning or practice; andApply reflective skills in practice.

Learning activities were specified for three areas of focus based on the work of Ash and Clayton [43]: personal development, professional development, and professional identity formation. These focus areas and associated activities were chosen to correspond to various purposes for reflection and different formats (e.g., written, verbal). A reflective rubric with seven elements of reflection, ranging from non-critical to critical, was selected to provide an evidence-based standard assessment strategy for instructors and students [42,44,45].

### 3.3. Development

The development phase involved production and evaluation of assignment templates for personal development (reflection on critical incidents) [8,46], professional development (reflection on learning) [47], and professional identity formation (reflection on practice experiences) [48]. Each template provided background information to explain the focus, objectives, guiding questions, format, and assessment of the reflection. These generic and modifiable templates were intended to be flexible, allowing for possible customizing of the assignment by instructors. These templates were also intended to be made available to students to support their own engagement in the PharmD program and beyond.

#### 3.3.1. Target Audience Review

All reflective activities were evaluated in the development phase via target audience review. The objectives of the target audience review were to explore students’ initial reactions to the assignment templates and document their perceptions regarding the clarity of instructions, purpose of the assignment, understanding of different types of assignments, and willingness to complete the assignment. Cognitive interviews [49] were conducted (KH) with a purposive sample of students who had completed years 1–3 of the PharmD program [50]. Prospective participants were identified based on the following criteria: availability, interest in reflection, and willingness to provide feedback. Initially, 14 students were contacted via social media to participate, with eight offering their time. The assignment templates were sent to these students 1 h before their interviews, which were conducted remotely using Zoom due to the COVID-19 pandemic. At the start of each interview, descriptions of the project and purpose of the interview were provided. Procedures to obtain informed consent were followed, and students were informed that their participation was voluntary, they could end the interview at any time, their comments would be anonymous and confidential, and their comments would be used to make improvements to the templates. Each student reviewed two of the assignment templates, which were randomly selected by the interviewer (KH). The cognitive interview involved asking students to share their thoughts, reactions to, and feelings about the assignments. At the end of each interview, the students were invited to ask the interviewer any further questions they had about the assignment templates. Research Ethics Board approval was not required for this aspect of the evaluation since the intent was to inform about development of instructional materials (S. Varnhagen, personal communication, 12 May 2020).

In total, eight interviews that were 30–45 min in duration were held over 4 days in May 2020. Interviews were not recorded. Notes were typed directly onto the assignment document by KH as the students talked about the assignment templates. Representative evaluation data is provided in Table 1.

Students’ initial reactions toward the reflection assignment templates were positive. They expressed appreciation for inclusion of specific objectives and guiding questions they felt would help students direct their learning and recognize different areas of focus for reflection. This appreciation for specificity also extended to the rubric as a guide for expectations. One student commented, “Even if a student wanted to try to fake their way through this assignment, they would still be learning something because the points on the rubric are so specific to hit.”

Some students had difficulty differentiating between the assignments involving reflection on personal development versus reflection on professional development. This concern was expressed by first-year students who stated that they had not been introduced to the concept of continuing professional development [47]. Students in years 2 and 3 were able to identify and explain the difference.

All students favoured the use of formative assessment over summative assessment. One student questioned whether it was appropriate to assign grades based on a student’s personal thoughts and experiences. Students expressed appreciation for the assessment rubric chosen for the assignments [44,45]. In terms of improvements, some students suggested revisions, as they found the assessment rubric too dense.

Students indicated that sharing their reflections with peers enhanced their learning. However, in terms of providing peer assessment for fellow students, there was a mixed response. Students supported the use of peer feedback in situations when there was no grade or risk associated with it.

#### 3.3.2. Expert Review

All reflective activities were also evaluated in the development phase via expert review. Pharmacy educators/researchers with expertise in reflection and innovative instructional approaches to teach patient care skills (CL, TLC) performed expert reviews of the reflection assignment templates. They suggested elaborating on the areas of focus by including references and definitions of critical incident, continuing professional development, and professional identity. They also suggested that students be provided with opportunities to reflect on both positive and negative experiences.

#### 3.3.3. Reflection Assignment Templates

Following the target audience and expert reviews, the reflection assignment templates (Appendix A), and assessment rubric (Appendix B) were modified. A possible reflective practice framework incorporating these elements was outlined to guide implementation across the curricular streams in year 1 (Table 2).

This framework introduces students to reflection in the three previously mentioned different areas of focus: personal development, professional development, and professional identity formation. In year 1 of the PharmD program at U of A, foundational concepts are taught in four of the six curricular streams (BASE, Patient Care Skills, Interprofessional Education, and Experiential Education). Including social aspects of reflective practice exposes students to others’ stories and thoughts, increasing opportunities for learning [19,20,41,51]. The first assignment in the fall term requires reflection on a critical incident [46]. This first reflection is a formative assessment involving a standard rubric and feedback on the reflective process. In the winter term, the focus becomes professional development [47]; both formative and summative assessment could be incorporated using the standard rubric. Reflective activities co-occurring with experiential education placements in the spring/summer term focus on professional identity formation [48]. In all three terms, the assignment format varies; students complete formal written reflections, continuing professional development plans, and informal writing in the form of a blog entry.

In subsequent years, reflection assignments increase in number. Activities vary, and reflection is used for different purposes (e.g., interprofessional collaboration, social justice projects), assigned in different formats (e.g., portfolios, podcasts, exhibits, stories), and assessed using different strategies (e.g., self, peer) to support further development of reflective practice abilities [9,22,52].

### 3.4. Implementation

Reflection assignment templates were implemented on a trial basis in a year 3 patient care skills course delivered remotely in the Fall 2020 term in conjunction with an evolving effort to develop care planning skills [53] and explore aspects of reflective practice [9]. The course, PHARM 420, is delivered weekly beginning with a one-hour seminar followed by three hours of skills development via simulation and small-group learning. Students were introduced to the reflection assignment during the pre-lab seminar. The assignment required each student to submit a written reflection following the session on a critical incident related to a disagreement with a physician (Appendix C). All 124 students enrolled in the course submitted their reflection assignments. The assignments were graded by the lead instructor (RRB). The assessment rubric was integrated into the content management system, eClass, to facilitate formative feedback on this assignment. Students received a completion grade of 1%.

### 3.5. Evaluation

The instructor’s evaluation of the reflection assignment facilitated identification of areas to improve its implementation in this course. While the grading process was essential to provide feedback to the students and assess students’ reflective practice ability, grading required several hours of instructor time. The assignments were well done overall; however, a wide variation in students’ reflective writing abilities was evident. Based on this experience, adjustments for the Fall 2021 course included changing the format to a small-group activity to simulate reflection-in-practice [9], combining group reflection discussion and writing in the assignment. An additional guiding question prompted students to explore the value of the group reflection and address any disagreement (Appendix C). Following grading of the group submissions, the instructor discussed the activity with all students at the seminar the following week and provided additional feedback verbally in a large group setting. The flexible reflection assignment template allowed the instructor to customize reflective practice activities for this course.

As a result of the implementation of these reflections in the PHARM 420 course, the templates are currently being evaluated and further modified. Future enhancements under consideration include peer assessment and feedback and the use of an online tool to support self-assessment [54].

## 4. Discussion

Researchers and educators have identified numerous challenges in teaching reflective practice skills, highlighting the need for a structured framework for reflection in health education [19,22]. Lack of understanding of or engagement in reflection on the part of students has long challenged educators [12,13,16,25]. Applying the concept of co-creation [27], a student-faculty partnership was formed at the University of Alberta’s Faculty of Pharmacy and Pharmaceutical Sciences, College of Health Sciences to develop reflection assignment templates. These templates offer guidance and flexibility for instructors to customize reflection assignments for pharmacy students across the curriculum. A previously tested, reliable assessment rubric is applicable to a broad range of settings [44,45]. Initial implementation showed promise, but further development is planned and broader implementation in the PharmD program is anticipated.

Initial evaluation of the templates in the PHARM 420 course encouraged continued use of the rubric to establish specific expectations for reflection and provide feedback to students. Although students questioned the value of graded reflection assignments, there may be situations when summative assessment is appropriate. The tested rubric provides guidance for summative assessment as it utilizes an objective process so that a grade is not related to the student’s thoughts, feelings or reactions but related to the fact they had initial thoughts, feelings, and reactions which they would reflect on to change future behaviour [44,45].

Students can learn much about reflective practice through their interactions with various assessment strategies [42]. Peer assessment may offer an impactful approach to enhance student engagement with reflection assignments, as sharing reflections with peers enhances learning [13,19,25,41,55,56]. Implementing self-assessment and peer feedback utilizing the rubric may be feasible for students in years 2 and beyond who have sufficient experience developing their own reflective practice abilities. Further research on the use of peer- and self-assessment in pharmacy education is warranted.

The co-creation process was critical to the outcome of this project. Student voice in curricular development and design of educational materials is imperative for continued improvement of instruction in reflective practice. Students bring a unique viewpoint to the problem, improving our understanding of what students need and why they accept or reject learning activities. Partnering with students who are passionate and willing to improve their understanding of reflective practice can result in development of engaging instructional techniques and materials [57]. Fostering relationships between students and professors as co-creators of the curriculum can give them both insight into each other’s challenges, allowing for growth, understanding, and empathy [33,36,57,58].

Co-creation projects can also promote further development of meta-cognitive abilities and transformative learning [33], as is evident in this reflection (KH):
*As a student who has never attempted creating educational materials, this was a very eye-opening experience. Now I can understand the work and time that has to go into creating these assignments; it is not as simple as writing up a task with a rubric. My primary goal going into this project was to increase the positive perception and understanding of reflective practices for the next generation of professionals. Was I able to achieve this? Yes, and no. I feel I was able to increase the positive notion behind reflection among the students who were part of the project. However, they make up an incredibly small portion of our student body. I also feel I was able to increase my own understanding of reflection and reflective practices through reading and analyzing journal articles.*
*The major change of heart (my own critical incident) I experienced during this project occurred when students reviewed the reflection assignment template for personal development with the option to share a reflection with a peer. My assumption was that students would embrace the opportunity to “save face” and remain anonymous when sharing their reflections and experiences with other students. Students loved the idea of discussion boards to anonymously post reflections and learning experiences with one another. However, when it came to the peer feedback, students stated they would prefer the reviewers be identified. Students related this to what they have learned in the PharmD program. They recalled that during lectures and learning to give feedback, they were taught the CORBS model [59]. The “O” stands for owned. Students stated commitment to best practices of feedback and to be honest with who is providing the review. My perspective changed because of this experience. I originally thought students would prefer anonymous feedback. My assumptions were reinforced by what I was reading in the literature. I see now this is not the case. I have a new appreciation for seeking input from students representing the target audience for the learning activity.*

For the student involved in the co-creation process, there were several experiences that could not be attained in a classroom. Originally, this project was intended to promote a shared understanding of the necessity of reflective practice for professional growth. However, the different aspects of the co-creation process increased the student’s passion and encouraged the use of reflection for the purpose of growth. Additionally, the project incorporated aspects of instructional design, research, and academic writing. With no previous experience in these areas, the student found that it was an incredible opportunity to be immersed in this kind of work, even with all the emotional ups and downs involved in writing, editing, reviewing, and submitting an article for publication. These realities are acknowledged within the academic world, but it is difficult to truly understand until you are placed in the same position [60]. Finally, the project fostered several positive relationships among faculty members and students. In this directed studies course project, where a power hierarchy is inevitable [36], the student (KH) received enthusiastic support. While co-creation typically aims to produce positive effects for students directly involved in co-creation, engagement by more students has potential for greater impact on students’ experiences [35].

Given the breadth of research and literature on reflection and reflective practice in a variety of disciplines, one limitation of this project is that some teaching and learning techniques may have been overlooked. This limitation was mitigated somewhat by the expertise of the individuals involved in the project. The co-creation process primarily involved one student and eight other students contributing to aspects of the development phase of the project. With respect to student voice, the purposive sampling method in the development phase did not represent all student perspectives; therefore, bias associated with the purposive sampling method may have influenced results. Future co-creation projects may benefit from involvement of larger groups of students [36]. Another limitation relates to the lack of representation from year 4 students and pharmacy graduates who may have valuable insights into what reflective practice means to them now that they have completed their entry-to-practice degrees. Despite these limitations, the cognitive interview method successfully gleaned specific and detailed information about what matters to students. Their insights enhanced the reflection assignment templates and assessment rubric. Further development, implementation, and evaluation in future will provide additional data based on views and experiences of students and instructors using the assignment templates, as well as the impact of co-creation on the learning environment, motivation of learners, and quality of the education design [35].

## 5. Conclusions

This project applied a co-creation process to develop reflective practice activities for pharmacy students using the ADDIE instructional design framework. Assignment templates were developed in three specific areas of focus for reflection: personal development, professional development, and professional identity formation. Initial implementation in a patient care skills course reinforced the flexibility of the templates that can be customized in future to meet course goals and student needs. The co-creation process enriched the experiences of students and faculty involved in the project. Future projects may include larger groups of students and formal evaluation of student and instructor experience as well as the impact of co-creation on the program.

## Figures and Tables

**Table 1 pharmacy-10-00028-t001:** Representative data from the target audience review.

Focus	What Works Well?	What Could Be Improved?
Reflection for personal development	500 words is an appropriate amount for a reflection. It allows for just enough filler to provide background information. (Year 2 Student) I really enjoy the prompting questions. I could think of an example to reflect on as soon as I read these questions! (Year 1 Student)	Change the wording from “critical incident” to “significant incident.”(Year 1 Student) I don’t understand the meaning of “appropriation” of knowledge [in the rubric]. Is that not the same as “integrating”? (Year 1 Student)
Reflection for professional development	I like having no numerical value/grade tied to the rubric. I do not believe students should be assigned a grade on reflections. How can a grader value one person’s thoughts more superior to another? (Year 3 Student) I like the variations that allow for peer feedback. I know I would personally put more effort into my work if I knew my colleagues would be looking at it. (Year 2 Student)	I think asking us [as first-year students] to complete this assignment in first semester might be difficult as we are just beginning to grasp concepts. (Year 1 Student) I think the student could also reflect on what the pharmacist is like. Are they friendly or jaded? Overworked or bubbling with energy? (Year 2 Student)
Reflection for professional identity formation	Public speaking is a great way to integrate other skills we need to have as a professional and allows for a good tie in with PHARM 454 [experiential education—hospital placement]. (Year 2 Student) I like how this rubric is applicable across all three assignments so we can integrate the aspects we should be touching on when reflecting. However, the rubric is quite wordy, it could be simplified. (Year 3 Student)	Students should be provided examples of reflection from their professors and how it has driven their practice to provide us [students] with a good reason to use reflection in the future. (Year 2 Student) I wish the assignments were not so fragmented [in the curriculum]. I wish we could have done a reflective portfolio that could have been tracked across our four years of pharmacy so we could see how we have grown. (Year 3 Student)

**Table 2 pharmacy-10-00028-t002:** Possible framework for reflective activities in year 1.

	Fall Term	Winter Term	Spring/Summer Term
Focus	Personal development	Professional development	Professional identity formation
Concepts taught in curricular streams	Reflective practice, reflection, critical incident	Learning, continuing professional development	Role of pharmacists, professional identity
Social	Student “near peer” shares experiences developing reflective practice	Discussion with peers	Sharing with peers
Format	Reflective writing (Critical incident)	Continuing professional development plan (Learning needs)	Blog entry (Practice experience)
Assessment	Formative	Formative, Summative	Formative

## Data Availability

Not applicable.

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
