# Peer review of "Reflective Practice: Co-Creating Reflective Activities for Pharmacy Students"

_pharmacy, 2022, doi:10.3390/pharmacy10010028_

Round 1
Reviewer 1 Report
I find it very interesting and inspiring that there has been growing interest in students and faculty working together to create curricula in recent years. Also, I believe that the topic is very important; describing co-creation process for the development of reflection assignments for pharmacy students so as to inform other pharmacy schools about the process.
However, the results section is not logical. Although the aim of this project was to co-create reflection assignments for pharmacy students and the methods section nicely follows this aim by describing the process, the results seem unrelated to the methods. The results merely touch upon implementation and evaluation. If these are the main results than the aim should be changed.
Reviewer 2 Report
See attached file

Round 2
Reviewer 2 Report
N/A